# Understanding Marine Biodegradation of Bio-Based Oligoesters and Plasticizers

**DOI:** 10.3390/polym15061536

**Published:** 2023-03-20

**Authors:** Federico Zappaterra, Monia Renzi, Manuela Piccardo, Mariachiara Spennato, Fioretta Asaro, Martino Di Serio, Rosa Vitiello, Rosa Turco, Anamaria Todea, Lucia Gardossi

**Affiliations:** 1Department of Chemical and Pharmaceutical Sciences, University of Trieste, Via L. Giorgieri 1, 34127 Trieste, Italy; 2Department of Life Sciences, University of Trieste, Via L. Girgieri 10, 34127 Trieste, Italy; 3Department of Chemical Sciences, University of Naples Federico II, Complesso Universitario di Monte Sant’Angelo, 80126 Napoli, Italy; 4Institute for Polymers, Composites and Biomaterials, National Council of Research, Via Campi Flegrei 34, 80078 Pozzuoli, Italy

**Keywords:** bio-based polyesters, bioplasticizers, marine biodegradation, biocatalysis, epoxidized oil, epoxidized fatty acids, cardoon oil

## Abstract

The study reports the enzymatic synthesis of bio-based oligoesters and chemo-enzymatic processes for obtaining epoxidized bioplasticizers and biolubricants starting from cardoon seed oil. All of the molecules had M_W_ below 1000 g mol^−1^ and were analyzed in terms of marine biodegradation. The data shed light on the effects of the chemical structure, chemical bond lability, thermal behavior, and water solubility on biodegradation. Moreover, the analysis of the biodegradation of the building blocks that constituted the different bio-based products allowed us to distinguish between different chemical and physicochemical factors. These hints are of major importance for the rational eco-design of new benign bio-based products. Overall, the high lability of ester bonds was confirmed, along with the negligible effect of the presence of epoxy rings on triglyceride structures. The biodegradation data clearly indicated that the monomers/building blocks undergo a much slower process of abiotic or biotic transformations, potentially leading to accumulation. Therefore, the simple analysis of the erosion, hydrolysis, or visual/chemical disappearance of the chemical products or plastic is not sufficient, but ecotoxicity studies on the effects of such small molecules are of major importance. The use of natural feedstocks, such as vegetable seed oils and their derivatives, allows the minimization of these risks, because microorganisms have evolved enzymes and metabolic pathways for processing such natural molecules.

## 1. Introduction

There is a debate around the environmental superiority of bio-based polymers and chemicals as compared to their fossil-based counterparts. Most life-cycle analyses show that bio-based plastics have a much lower upstream impact compared to their oil-based equivalents [1]. This appears quite evident when considering that in 2019 plastics generated 1.8 billion tons of greenhouse gas (GHG) emissions—3.4% of global emissions—with 90% of these emissions coming from their production and conversion from fossil fuels. Notably, in 2021, global production of plastics rose by 4% to more than 390 million tons, demonstrating the strong and continuing demand for plastics. The estimated global leakage of plastics to the environment (terrestrial and aquatic) was 22 Mt in 2019. This value is projected to double, reaching 44 Mt by 2060 [2]. Nevertheless, fossil-based plastics still represent more than 90% of global plastic production, while post-consumer recycled plastics amount to about 8.3%, and bio-based plastics account for only 1.5% of the total production [3].

Plastic debris represents a major danger for the majority of marine species. On the other hand, the development of novel biotechnological approaches for the sustainable biological degradation of recalcitrant plastic needs to start from the rational eco-design of the polymer. Indeed, the aspects that influence the degradation rate of polyester are numerous, especially in marine environments [4]. Intrinsic factors of the polymer (such as its chemical structure, crystallinity, molecular weight, shape, etc.) have the greatest effect on the degradation process [5]. Moreover, the degradation behavior of polymers in marine environments is also due to characteristics of seawater that influence the rate of degradation (such as types and numbers of microorganisms, temperature, UV exposure, pH, and salinity in different waters) [6]. Among these external environmental factors, the temperature and different microbial communities remain the most critical determinants of the degradation rate of polyesters [7].

Concerning the end-of-life and the fate of bio-based polymers dispersed in the environment, there are bio-based polymers—such as bio-polyethylene or bio-polyamides—that are designed for being durable, since biodegradability is not a desirable property for certain applications, e.g., in the automotive or textile sectors [8,9]. Therefore, the misuse of such durable bio-based polymers might lead to downstream environmental impacts because biodegradation does not depend on the resource basis of a material, but rather on its chemical structure [10].

However, there are bio-based chemicals—especially some polyesters—for which biodegradability is a desired property. An example is represented by the polymeric ingredients applied in a variety of cosmetic formulations or sunscreen lotions that are poured down the drain after use or directly released into the sea. Other examples include lubricants used in machinery operating in ecologically sensitive environments, such as marine (e.g., workboats and passenger boats) and agricultural areas [11,12].

A third case is represented by plasticizers that are added to plastics during their manufacture to improve their mechanical and thermal properties [13]. When the environmental impacts associated with plastic were calculated via life-cycle analysis (LCA) techniques, using official databases such as the US Toxic Release Inventory [14] and Plastics Europe eco-profiles [15], the impact of additives’ leachate from plastics was also considered. A study calculated the amount of additives per type of plastic, based on an OECD report [16], finding that the annual leaching rate of additives is 0.16% per year, which means that it would take 625 years for 100% of the additives to be released from the plastics. Conversely, the fate and toxicity of these additives is of major importance when designing new types of plastic formulations [17,18].

The integration of chemistry and biocatalysis enables the delivery of an array of bio-based products designed for decreasing the hazards derived from certain polymers and chemicals that, because of their specific applications, are likely to be dispersed in the environment [19]. Most importantly, these bio-based polymers and chemicals avoid GHG emissions connected to the extraction and processing of fossil-based feedstocks [20].

In the present study, two bio-based oligoesters with potential applications in the dermatology and cosmetics sector were synthetized by enzymatic polycondensation to control the length and structure of the products. The bioplasticizers were obtained by chemical epoxidation of cardoon seed oil, which was also transformed via epoxy-ring opening in the presence of different alcohols and polyols to tune their properties as plasticizers or lubricants. More specifically, the investigation included not only the bio-based chemicals, but also their precursors. All molecules considered in the present study—both natural and synthetized—are characterized by the presence of ester bonds and have M_W_ below 1000 g mol^−1^. Therefore the objective of the investigation was not in to observe the erosion of the surface of a polymer/plastic debris but, rather, to focus the attention at the molecular and chemical levels and to observe two different phenomena: (i) the initial phase of the biodegradation involving the breaking of the most labile bonds, generally corresponding to the hydrolysis of the ester bonds or the opening of the epoxy rings; and (ii) the biodegradation of the single monomers and some model epoxidized fatty acids obtained via chemo-enzymatic routes. The combination of these two distinct types of biodegradation information provides useful insights on how not only the specific structural components but also the nature of the chemical connection of the building blocks affects the biodegradability of each bio-based product. The results presented herein shed light on some fundamental chemical features that affect the biodegradability of the bio-based products under investigation, by distinguishing between the disassembly of the building blocks and their actual biodegradation.

## 2. Materials and Methods

### 2.1. Materials

For the synthesis of the oligoesters, the following materials were used: Lipase B from *Candida antarctica* (CaLB, EC 3.1.1.3, specific enzymatic activity = 368 TBU/g, immobilized on epoxy-acrylic resins), 1,4-butanediol (CAS No. 110-63-4, purity = 99%), glycerol (CAS No. 56-81-5, purity = 99,5%), adipic acid (CAS No. 124-04-9, purity > 99%), dichloromethane (CAS No. 75-09-2, purity > 99.9%), hydrogen peroxide solution (CAS No. 7722-84-1, concentration = 30%), deuterated chloroform (CAS No. 865-49-6, purity = 99.8%), oleic acid (CAS No. 112-80-1, purity = 99%), linoleic acid (CAS No. 60-33-3, purity = 99%), and linolenic acid (CAS No. 463-40-1, purity = 70%) were supplied by Sigma-Aldrich (Milano, Italy)-). Azelaic acid (CAS No. 123-99-9, purity = 98%) and cardoon oil, with iodine values of 125 (gI_2_/100gsample), were kindly donated by Novamont S.p.A. (Novara, Italy). Hydrogen peroxide (60 wt.%) was kindly provided by Solvay Italia (Bollate, Italy). All of the other chemicals were purchased from Merck and Sigma (Milano, Italy) (analytical or reagent grade), and they were used as received.

### 2.2. Methods

#### 2.2.1. Enzymatic Activity Assay

The hydrolytic activity of the tested lipases was evaluated using tributyrin, as previously described by Spennato et al. [21]. An emulsion composed of 1.5 mL of tributyrin, 5.1 mL of arabic gum emulsifier (0.6% *w*/*v*), and 23.4 mL of water was prepared to obtain a final molarity of tributyrin of 0.17 M. Then, 2 mL of K-phosphate buffer (0.1 M, pH 7.0) was added to 30 mL of tributyrin emulsion, and the mixture was incubated in a thermostatted vessel at 30 °C, equipped with a mechanical stirrer. After pH stabilization, 50 mg of biocatalyst was added. The consumption of 0.1 M sodium hydroxide was monitored for 15–20 min. One unit of activity was defined as the amount of immobilized enzyme required to produce 1 mmol of butyric acid per min at 30 °C.

#### 2.2.2. Enzymatic Synthesis of 1,4-Butandiol- and Glycerol-Based Oligoesters

In a 100 mL flask, adipic acid/azelaic acid and 1,4-butanediol/glycerol were added at a molar ratio of 1:1. The reaction was started by the addition of the immobilized enzyme CaLB (128 enzymatic units per g of total monomers). This reaction can be defined as solventless because 1,4-butandiol and glycerol act both as reagents and as reaction media [22,23]. The reactions were performed under vacuum in a rotary evaporator at 70 °C and 70 mbar for 72 h. At the end of reaction, the product was recovered using dichloromethane, and the solvent was removed by vacuum-drying at 40 °C.

#### 2.2.3. Chemo-Enzymatic Epoxidation of Fatty Acids in Solventless Conditions

The reactions were performed on a 2 g scale of fatty acids at 50 °C in the presence of immobilized lipase B from *Candida antarctica* (CaLB Novozyme 435), commercialized by Novozymes (Denmark), with a specific activity of 1998 U/gdry according to the tributyrin hydrolysis assay. The biocatalyst (225 U per g of oleic acid and 450 U per g of substrate in the case of linoleic and linolenic acids, respectively) was added to the fatty acids and mixed in a 25 mL round flask for 15 min in a water bath at 50 °C. After this time, H_2_O_2_ was added to the reaction system (stepwise additions) at a molar ratio of 2:1 in terms of the C=C bonds. The reactions were performed at 50 °C in a water bath. At the end of the reaction (2 h for oleic acid, 3 h for linoleic and linolenic acids), the product was extracted using dichloromethane (3 × 20 mL DCM). Then, the organic phase was recovered and dried with sodium sulfate, and the mixture was stirred for 30 min. Sodium sulfate was filtered through a Buchner funnel with filter paper, and the dichloromethane was evaporated on a rotary evaporator while keeping the temperature below 40 °C. The conversion of C=C into epoxy was calculated by ^1^H NMR. The products were also analyzed by GC-MS.

#### 2.2.4. Characterization of Epoxidized Fatty Acids by GC-MS Analysis

For GC-MS analysis, the derivation of compounds was performed by using bis(trimethylsilyl)trifluoroacetamide (BSTFA) according to the protocols reported in the literature [24]. The GC-MS samples were prepared by dissolving 1 mg of compound in 1 mL of toluene. GC-MS analysis was carried out using a Shimadzu gas chromatograph equipped with a 30 m × 0.25 mm fused-silica capillary column (SLB5ms) coated with a 0.25 µm film of poly(5% phenyl,95% dimethyl siloxane). The temperature was monitored from 100 °C to 300 °C. Dodecane was used as an internal standard.

#### 2.2.5. Synthesis of ECO Sample

The epoxidation of cardoon oil was carried out in a jacketed glass reactor (500 mL) equipped with a thermocouple, a reflux condenser, and a mechanical stirrer (750 rpm), according to the experimental procedure reported in detail in [25]. More specifically, 100 g of oil was epoxidized with performic acid produced in situ by a reaction between hydrogen peroxide (37 g) and formic acid (5.38) in the presence of orthophosphoric acid (1.2 g). The reaction mixture was stirred at 70 °C for 3 h. At the end of the reaction, the product was withdrawn from the reactor, quenched, and centrifuged at 3000 rpm for 20 min. The final products were analyzed by evaluation of the iodine number (I.N.) following the Wijs method [25] and the oxirane number (O.N.) according to the standard method ASTM D1652–11 [26].

#### 2.2.6. Synthesis of ECO-BDO and ECO-SRB Samples

The epoxidized cardoon oil, as described above, was modified by reaction with polyalcohols—such as 1,4-butandiol and sorbitol—through an acid-catalyzed ring-opening mechanism to obtain polyols, as shown in Appendix A. The ring-opening reactions were conducted in a 250 mL magnetic glass stirred reactor immersed in an oil bath and thermostatted with an electrical heating device, equipped with (i) a glass thermometer for measuring the reaction temperature, (ii) a vertical bubble condenser to prevent any loss of the alcohol by evaporation, and (iii) a sample withdrawal for chemical analysis. All of the experimental runs were conducted at 120 °C for three hours. The typical experimental procedure was as follows: Epoxidized oil and alcohol (1,4-butandiol or sorbitol), at a 1:1.5 molar ratio, were weighed and loaded in the reactor, and the stirring and heating were started; thus, the reaction mixture gradually reached the desired reaction temperature. At time zero, sulfuric acid, used as a catalyst (2 wt%), was added to the reactor. At the end of the reaction, the product (a single homogeneous phase with an intense yellow color) was quenched and neutralized with a solution of ammonia (NH_3_) at 30% by weight, and then it was centrifuged and characterized by determination of the oxirane number and hydroxyl number [26].

#### 2.2.7. NMR Analysis

Up to 20 mg of sample was weighed in a 4 mL screw-cap glass vial, and 800 µL of deuterated solvent (either DMSO-d_6_ or CDCl_3_) was added to achieve complete dissolution. The NMR measurements were carried out on a Varian VNMRS-500 (11.74 T) and on a Varian 400-MR NMR spectrometer (9.4 T), operating for protons at 500 MHz and 400 MHz, respectively.

#### 2.2.8. Thermal Analysis

The thermogravimetric analysis of the samples was performed using a TGA Q500 V6.3 Build 189 under a nitrogen atmosphere, with a heating rate of 10 °C min^−1^ in the temperature range 20–1000 °C. DSC analyses were performed using the DSC 300 Caliris differential scanning calorimeter (Netzsch, Selb, Germany) under a nitrogen atmosphere, in the temperature range −50–180 °C, at a heating rate of 10 K/min, in 2 heating/cooling cycles. The recorded data were processed with the Netzsch Proteus Thermal Analysis software version 9.0. (NETZSCH-Geraetebau GmbH, Selb, Germany).

#### 2.2.9. Biodegradation Studies

Biodegradation tests of the bio-based products, monomers, and vegetable oil were carried out in accordance with ISO 17556:2019 using the OxiTop^®^ Control S6 system, which used a respirometric method to measure the oxygen demand released during the aerobic biodegradation of organic materials—in our case, oligoesters. For the biochemical oxygen demand (BOD) measurements, the OECD 306 protocols and the OxiTop^®^ system were used. The OxiTop^®^ Control S6 system was equipped with six measuring units (amber glass bottles (510 mL) and self-check measuring units), an inductive stirring platform, and magnetic stirrer bars. The following components were added in each unit: 327.5 mL of the salt solution (prepared as described below), 1 mL of DMSO or sample dissolved in DMSO, and 36.5 mL of seawater (inoculum). The epoxy–oligoester mixture was dissolved in DMSO (36.5 mg/mL), and the sample was diluted with the salt solution to reach a final concentration of 100 mg/L. The bottles were closed with the measuring cap and were placed in an incubator with a controlled temperature of 21 ± 1 °C. The salt solution was prepared by diluting 1 mL of each solution 1 ÷ 4 with distilled water to reach a final volume of 1 L. *Solution 1* was prepared by dissolving 8.5 g of potassium dihydrogen phosphate (KH_2_PO_4_), 21.75 g of dipotassium hydrogen phosphate (K_2_HPO_4_), 33.4 g of disodium hydrogen phosphate heptahydrate (Na_2_HPO_4_*7H_2_O), and 1.7 g of ammonium chloride (NH_4_Cl) with distilled water to a final volume of 1 L. *Solution 2* was prepared by dissolving 22.5 g of magnesium sulfate heptahydrate (MgSO_4_ · 7H_2_O) in distilled water to a final volume of 1 L. *Solution 3* was prepared by dissolving 27.5 g of anhydrous calcium chloride (CaCl_2_) (or an equivalent amount if the hydrate was used, e.g., 36.4 g CaCl_2_ · 2H_2_O) in water to a volume of 1000 mL and then mixing. *Solution 4* was prepared by dissolving 0.25 g of iron(III) chloride hexahydrate (FeCl_3_ · 6H_2_O) in water to a final volume of 1 L. The biochemical oxygen demand (BOD) for each sample was determined using the following equation:(1)$$BOD_{s}\;=\;\frac{BOD_{x}-BOD_{g}}{c}$$
where *S* is the number of measurement days, *BODs* is the biochemical oxygen demand of the analyzed sample within *S* days (mg/L), *BOD_x_* is the biochemical oxygen demand of the measuring system (bottle with sample and water) (mg/L), *BOD_g_* is the biochemical oxygen demand of water without the sample (mg/L), and *c* is the sample concentration in the tested system (mg/L).

The degree of biodegradation of the product was determined based on the following equation:(2)$$D_{t}\;=\;\frac{BOD_{s}\;}{TOD}\times 100$$
where *D_t_* is the degree of product degradation (%), while *TOD* is the theoretical oxygen demand (mg/L). The theoretical oxygen demand for each system was calculated using the following equation:(3)$$TOD\;=\;\frac{16\times \left(2\times C+0.5\times H-O\right)\;}{M_{n}}\times 100$$
where *C*, *H*, and *O* are the mass shares of elements in the molecules of the biodegradable material, while *Mn* is the average molecular weight of the biodegradable material

## 3. Results and Discussion

### 3.1. Enzymatic Synthesis and Characterization of Oligoesters

The bio-based oligoesters poly(butyleneadipate) and poly(glycerolazelate) were investigated to understand some of the drivers of marine biodegradation in bio-based polyesters, which are not natural polymers but are obtainable through the enzyme-catalyzed synthesis of ester bonds connecting the bio-based monomers. The use of a lipase as a catalyst allowed for the control of the molecular weight and the structure of the products—for instance, avoiding branching [27].

Lipase B from *Candida antarctica* (CaLB) was employed for catalyzing the polycondensation of adipic acid and 1,4-BDO to obtain poly(butyleneadipate) (BDO-AA), as well as that of azelaic acid and glycerol to yield poly(glycerolazelate) (Gly-AZA), exploiting the specificity of CalB for the primary hydroxyl groups [28]. The process was carried out at 70 °C under solventless conditions and using a thin-film system, as previously described by Todea et al. [27].

The oligoesters were characterized by ESI MS spectrometry. The typical ESI MS spectra of the reaction products obtained in a solventless system in the presence of covalently immobilized CalB are presented in the Appendix A. The structural characterization of the reaction products was also performed via NMR spectroscopy, and the NMR spectra are presented in the Appendix A).

These oligoesters were selected not only for their potential industrial relevance, but also because they have already been the object of extensive investigations concerning their synthesis, structural properties, and degradation. Previously, we have reported the synthesis of oligoesters of poly(glycerolazelate) [27] for potential use in dermatological and cosmetic formulations, due to the pharmacological properties of azelaic acid [29]. Therefore, these oligoesters are subjected to be washed off and collected in the wastewater, ultimately reaching the sea.

The fate of poly(butileneadipate) is similar, since it cannot be recovered or recycled, due to its final use as an ingredient of adhesive and coating formulations. Indeed, the synthesis of poly(butileneadipate) was the first enzymatic polycondensation carried out on the industrial scale, achieved in the 1990s by Baxenden Chemicals (UK) for the production—later dismissed—of highly regular structures of polymers used for these applications [30]. More recently, we investigated the same enzymatic polycondensation but using recyclable covalently immobilized enzymes in thin-film systems [31]—also on a pilot scale [32]—to overcome the technical limitations observed by Binns and co-workers [30]. Our previous studies on the enzymatic synthesis of both poly(butileneadipate) and poly(glycerolazelate) indicated that the processes are potentially scalable, since the biocatalyst can be efficiently removed after the synthesis of short prepolymers. As a matter of fact, even in the absence of the enzyme, the elongation of the polyesters can be effectively pushed thermodynamically, via the removal of the water produced in the esterification.

Notably, previous computational simulations shed light on the ability of different hydrolases to recognize poly(glycerolazelate) as a substrate in both synthetic and hydrolytic reactions [27]. As the synthesis of the ester bond is a reversible reaction, these oligoesters can in principle undergo hydrolytic degradation catalyzed by the same enzymes used for their synthesis—namely, triacylglycerol hydrolases, which are quite ubiquitous in nature. Furthermore, these oligoesters (Table 1) can be also considered as simplified models of polymers that are usable for formulating bio-based plastics. with low molecular weights Their low molecular weight (between 683 and 926 g mol^−1^) allows us to neglect factors such as the shape and thickness of samples, while focusing the attention solely on specific chemical and structural factors at the molecular level.

However, the translation of the potential biodegradability of a chemical product—whether natural or obtained by synthesis—into actual biodegradation depends strictly on the environmental conditions. The data presented herein provide information on the marine biodegradation of a set of bio-based products (chemical structures presented in Figure 1, Figure 2 and Figure 3), which are potentially biodegradable according to the OECD 306 protocols and under controlled conditions.

### 3.2. Synthesis and Characterization of Bioplasticizers and Biolubricants Derived from Cardoon Seed Oil

Plant oils are prone to different chemical modifications, such as transesterification, estolide formation, double-bond hydrogenation and epoxidation, subsequent ring opening and, finally, acylation of the resulting -OH groups [33,34]. Cardoons (*Cynara cardunculus* var. *Altilis*) are typical of the biomass of marginal and sub-arid Mediterranean territories, where this plant achieves its full production potential. Cardoon fruits are cypselae, usually indicated as cardoon seeds [35], which give oil with yields in the range of 14.5–32.4% of the dry matter mass fraction, depending on the extraction method [36]. This oil has a fatty acid profile similar to that of sunflower oil, with 11% palmitic, 4% stearic, 25% oleic, and 60% linoleic fatty acids [37]. Therefore, cardoon seed oil can be considered to be a sustainable, non-edible alternative for the purposes of biorefineries, since the plant can withstand severe drought and high soil salinity, with beneficial effects regarding soil properties, erodibility, and biological and landscape diversity [37]. Previous studies reported the chemical epoxidation of cardoon seed oil using γ-alumina as a heterogeneous solid catalyst in the presence of different solvents to obtain plasticizers used in biodegradable films made from poly(lactic acid) and thermoplastic starch [33,34].

In the present study the triglycerides were first modified at the level of the C=C bonds of the unsaturated fatty acids to create structural complexities that confer them with properties as either plasticizers or lubricants. The oil was epoxidized with performic acid produced in situ by a reaction between hydrogen peroxide and formic acid in presence of the orthophosphoric acid. The final product was analyzed to evaluate the iodine number (I.N.) according to the Wijs method [25] and the oxirane number (O.N.) according to standard method ASTM D1652–11 [26]. The results are reported in Table 2.

The epoxidized cardoon oil was then modified by reaction with 1,4-butandiol and sorbitol (Appendix A), through an acid-catalyzed ring-opening mechanism to obtain polyols, as shown in Figure 3.

The products were characterized in terms of oxirane number and hydroxyl number [26]. The results are reported in Table 2 and indicate the decrease in the oxirane number for both products. The iodine number (gI_2_/100g_sample_) was determined to be 4.3 for all three samples. The hydroxyl number values of ECO-SRB were affected by the lower reactivity of the secondary -OH groups of sorbitol. The decrease in the absorption bands in the FT-IR spectra (Appendix A) from 830 cm^−1^ confirmed the opening of the oxirane ring, and the important increase in the hydroxyl group’s absorption band intensity in the 3300–3400 cm^−1^ range confirmed the presence of OH groups in the products’ molecules.

All derivatives of cardoon seed oil were investigated to understand the extent to which the structural modification of a natural molecule (i.e., the triglyceride) modifies the marine biodegradability of its derivatives. Notably, the marine biodegradation tests also involved natural cardoon seed oil. Generally, the biodegradability of natural molecules cannot be taken for granted, since they are the results of biosynthetic processes. Therefore, it is generally assumed that nature provides degradation pathways to ensure the closure of the biogeochemical cycle of carbon. Nevertheless, there are various abundant natural materials, including lignin and rubber, which are quite resistant to biodegradation in most natural environments because of their function in nature, which requires high chemical stability and scarce reactivity of the bonds connecting their monomers [38]. As a consequence, they undergo biodegradation only under certain conditions and when attacked by specific enzymes, such as laccases in the case of lignin [39].

Conversely, we wanted to verify how the highly hydrophobic triglycerides are biodegraded in a diluted marine environment.

### 3.3. Chemo-Enzymatic Synthesis and Characterization of Epoxy Fatty Acids

In order to better understand the effect of the introduction of the epoxy rings on the biodegradation of the cardoon seed oil derivatives, oleic, linoleic, and linolenic acids were epoxidized using CaLB as a biocatalyst. In the enzymatic route, the formation of stable peroxy acids took place in situ by means of the enzymatically catalyzed reaction of H_2_O_2_ with the carboxylic group of free fatty acids, allowing for a significant suppression of side reactions.

Although CaLB is classified as a carboxylic ester hydrolase (3.1.1), it is endowed with perhydrolysis activity, since it replaces water with hydrogen peroxide as a nucleophile to attack the first tetrahedral transition state complex, forming an acyl–enzyme intermediate, and then releasing peracids [40]. Then, the peroxy acids formed in situ spontaneously react with the C=C, yielding the epoxide [41] through the Prilezhaev epoxidation mechanism for alkenes [42].

The different fatty acids were epoxidized (the chemical structures of the products are presented Figure 4) using the conditions reported in Table 3, and in all cases the epoxidation of the alkene bonds was >95%, as demonstrated by the disappearance of protons corresponding to the double bond in the ^1^H-NMR spectra (Appendix A) and the GC-MS analysis (Appendix A) available in ESI. Notably, the reactions were fast (2–3 h), and no other solvent was employed aside from 30% H_2_O_2_ solution. The recovery of the products was almost quantitative in all cases for an easy work-up.

### 3.4. Thermal Analysis of the Selected Compounds

The thermal properties of the oligoesters were evaluated by TG. The thermograms (Appendix A) indicate that except for the sample with sorbitol, the other molecules degraded in a single step, and the degradation started after 200 °C. Details about the mass loss at different temperature intervals are presented in Table 4. Among the tested molecules, the ECO-SRB showed a slightly lower thermal stability at temperatures of up to 300 °C. For all of the samples, more than 70% of the mass was lost at temperatures bellow 400 °C, which confirming the organic origin of the samples.

### 3.5. DSC Analysis

The use of differential scanning calorimetry (DSC) is relevant to deeply understand and correlate the effects of any type of degradation [43].

The DSC results obtained by cooling the samples from 20 to −50 °C and then heating them from −50 to 180 °C at the same scanning rate are reported in Figure 5, and further details are presented in Appendix A.

The values of the glass transition temperatures (T_g_), melting temperatures (T_m_), and melting enthalpies (ΔH_m_) determined from the areas of the melting peaks for the oligoesters are presented in Table 5.

For all of the samples—especially for the plasticizers and the azelaic acid oligoesters, which are viscous liquids at room temperature—the melting temperature was considered to be the last value of the peak obtained from the second heating cycle. The presence of different endothermic peaks on the DSC thermograms for the oil derivatives was previously studied by several groups and attributed to their different crystalline forms [44].

The melting behavior of BDO-AA—a semicrystalline polyester—indicates the presence of two closed endothermal peaks due to its polymorphic structure. This behavior was previously observed and studied in detail by Woo and Wu [45].

Among the plasticizers, the lowest melting point value was obtained for the product modified using sorbitol (ECO-SORB), followed by the samples modified with 1,4-butandiol (ECO-BDO) and with the epoxy oil.

### 3.6. Biodegradation Studies

To overcome several deficiencies of the biodegradation tests in real environments, controlled laboratory tests simulating aquatic environments are often used to evaluate biodegradation processes [46].

The biodegradation process was monitored using specific OxiTop^®^ devices (Xylem Analytics, Weilheim, Germany) that were equipped with sensors to measure the biochemical oxygen demand (BOD) required by aerobic microorganisms to degrade organic matter in each environment. The study was carried out in liquid culture media, in which the water was collected from the Adriatic Sea (Trieste) in the period November–December 2022 and was used as an inoculum. The experiments were carried out at 21 °C, and the biochemical oxygen demand was determined over 21 days, measured every 24 h. In addition to the aforementioned synthetized samples, the monomers used for the synthesis of co-oligoesters were also considered. The results, expressed as BOD (mg/L) and D_t_ (%) after 10 and 21 days, are summarized in Table 6. The complete values are presented in Figure 6 and Figure 7.

The results obtained for the tested monomers (Table 6, entries 1–3 and 10–12) indicate that after 21 days the compound with the lowest biodegradability was the epoxylinolenic acid (D_t21_ = 5%). On the other hand, the highest D_t_ value was obtained for adipic acid (D_t21_ = 35.44%). Even though glycerol is a non-toxic molecule—as previously reported by Wolfson et al. [47]—and the addition of glycerol in the structures of different polymers increased the biodegradability of the final products [48], in the marine environment the BOD_21 and D_t21_ values reached values of 55.95 mg/L and 10.55%, respectively.

The biodegradation trends recorded during the exposure time were not similar when comparing the tested chemicals. In some cases, over 50% of the total biodegradation occurred early and within 10 days of exposure. In fact, D_t21_ was less than 50% higher than D_t10_. The increase in biodegradation during the period D_t21_–D_t10_ ranged from 21.8% (EPX_LN) to 45.9 (BDO). A group of chemicals (BDO-AA, EPX_oleic, EPX_LNN) showed more than 68% of the total biodegradation occurring within D_t10_. Two chemicals (BDO and AA), although principally biodegraded within the first experimental time periods, showed that about 40–45% of biodegradation occurring in the second period of exposure.

The biodegradability data obtained with the oligoesters (entries 4 and 5) indicate that after 5 days the biodegradability of the oligoesters based on adipic acid and BDO was about twice as high as the D_t_ values of the oligoesters containing azelaic acid and glycerol. After 21 days, the ratio between the D_t_ values decreased to 1.36, and the adipic-acid-based oligoesters reached the value D_t21_ = 49.95%. This value is about 15% higher compared to the previous value reported by Kasuya et al. [49] for poly(butyleneadipate).

For the cardoon seed oil and its derivatives (entries 6–9), the highest D_t_ values were obtained for ECO-SORB and ECO-BDO—derivatives without double bonds or oxirane rings in the molecules, and probably with greater hydrophilicity compared to the starting oil or the epoxidized one. Interestingly, the biodegradation rate within the first 5 days was about 1.2-fold higher for the ECO-BDO sample compared to ECO-SORB, but later the order changed, and after 21 days the ECO-SORB sample reached a value of D_t21_ = 36.81%—about 1.33-fold higher compared to the ECO-BDO. The observed biodegradation behavior could be correlated with the T_m_ and T_g_ values determined by DSC and was consistent with the previously summarized data, where lower values of T_g_ and T_m_ were correlated with a higher degradation rate [43].

The degradation of the cardoon oil and the epoxidized oil (entries 8 and 9) was comparable; the values for the epoxidized sample were slightly higher after 21 days, but in the first 5 days the values were similar. These results indicate that the biodegradability is not affected by the presence of the oxirane rings.

The molecules with the lowest biodegradability were the epoxy fatty acids. The results indicated comparable values for epoxy oleic and epoxy linoleic acids, while the lowest values were for linoleic acid. The BOD values increased exponentially in the first 5 days, but later the biodegradation rate was much slower. These results could be attributed to the rigidity and low solubility of the molecules.

According to the US-EN ISO 14851:2019 guidelines, a substance can be considered biodegradable if the BOD is higher than 60% ThOD. Although longer degradation times than those reported in this study were proposed in the literature (i.e., 60–180 days, ISO 19679; ISO 23977-2; ASTM D7991-15), longer exposure times are also associated with an increased frequency of experimental failures due to the increasing difficulty in the execution of the test.

In some of the test cases, longer exposure times should be considered and tested, as the stability of the biodegradation curve was not completely reached during the exposure times tested (21 days) [50]. Marine water is typically slightly alkaline (pH = 8.1), so biodegradation of chemicals that occur more frequently in slightly acidic or neutral environments—such as esters and polyesters—is disadvantaged under such conditions, as they could induce a lack of the microbial strains needed in the marine medium to support effective biodegradation [51]. Notably, we have already demonstrated the lability of the ester between glycerol and azelaic acid that simulated the physiological skin environment at pH 4.5 [27].

With respect to the degradation trend during the experimental time, the temporal delay reported for AA in the degradation process could be due to the need for a preliminary reduction in the molecular weight to allow the microbial attack [52]. Recent research analyzed the biodegradation of PVA in marine environments, showing a role of the glycerol component in blended materials during the biodegradation process [48].

## 4. Conclusions

The present investigation reports on the synthesis of selected bio-based chemicals characterized by the presence of ester bonds and, in some cases, epoxy rings. All of the molecules, oligoesters, bioplasticizers, and biolubricants had M_W_ below 1000 g mol^−1^ and were analyzed in terms of marine biodegradation. Therefore, the objective was not the observation of the erosion of the surface of a polymer/plastic debris, but the determination of the effects of the chemical structure, chemical bond lability, and water-solubility on biodegradation. By also including in the marine biodegradation study the building blocks that are theoretically obtained upon ester bonds’ hydrolysis, their behavior was also observed, setting the basis for distinguishing between different chemical and physicochemical factors. These hints are of major importance for the rational eco-design of new benign bio-based products.

It is important to stress that the biodegradation results reported here should only be considered valid when referring to the specific inoculum used, geographical and seasonality variability, and the incubation temperature. Nevertheless, they allow a comparative view of the effects ascribable to the reactivity of chemical bonds and to the physicochemical properties of the bio-based molecules. It is known that abiotic and biotic processes proceed slowly and depend on a number of factors, including molecular weight, surface-to-volume ratio, and water-solubility. Overall our data confirm that the incorporation in the polymer of “weak links” [43]—such as ester bonds of aliphatic monomers—accelerates the abiotic degradation, and that polyesters—unlike polyolefins and polystyrene—also undergo enzymatic hydrolysis due to the enormous numbers of bacteria and microbial enzymes that are able to attack their ester bonds.

Nevertheless, the results presented here highlight that the simple analysis of the erosion, hydrolysis, or visual/chemical disappearance of the chemical products or plastic is not sufficient. The biodegradation data clearly indicate that the monomers/building blocks undergo a much slower process of abiotic or biotic transformations. Therefore, once the polymers or plasticizers of interest undergo biodegradation, an accumulation of other smaller chemical molecules occurs. In principle, the use of natural feedstocks—such as vegetable seed oil and their derivatives—can minimize these risks, not only because of their known low toxicity, but also because microorganisms have evolved enzymes and metabolic pathways for processing such natural molecules. Conversely, ecotoxicity studies on the effects of such small molecules are of major importance and will be the subject of our next investigation.

## Figures and Tables

**Figure 1 polymers-15-01536-f001:**
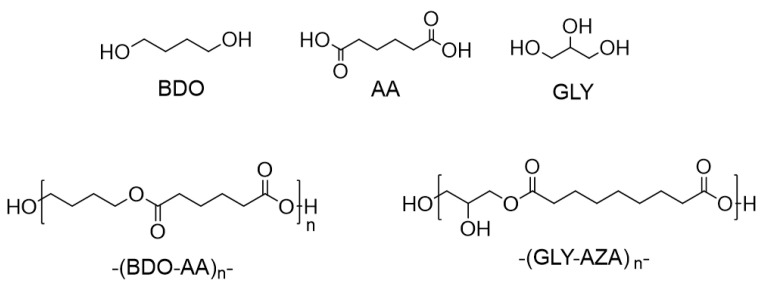
Structure of the bio-based monomers and the corresponding bio-based oligoesters obtained by enzymatic polycondensation. Codes refer to marine biodegradation tests.

**Figure 2 polymers-15-01536-f002:**
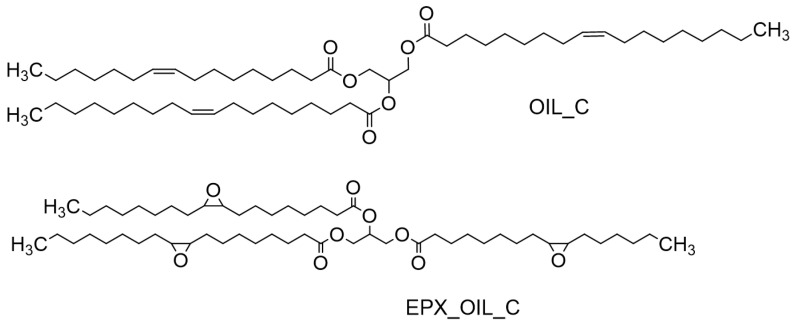
A schematic representation of the structure of an unsaturated triglyceride (triolein) and the corresponding epoxidized products. The cardoon seed oil is actually composed of a variety of saturated and unsaturated fatty acids, as reported above. The figure aims at illustrating the transformation of a triglyceride in a simplified way. Codes refer to marine biodegradation tests.

**Figure 3 polymers-15-01536-f003:**
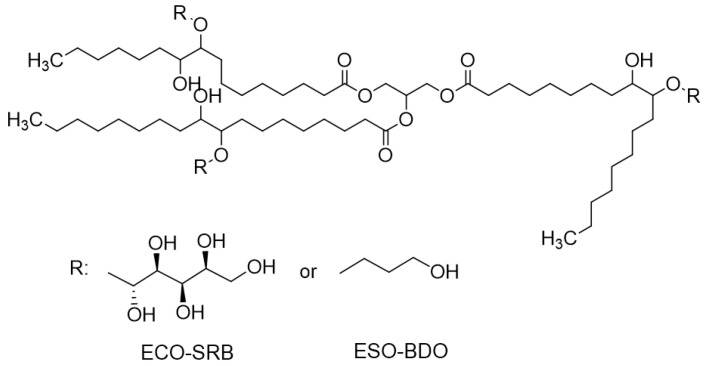
Schematic representation of products obtainable from the ring opening of the epoxidized triolein. As in the case of Figure 2, the structure does not represent the actual chemical complexity of the composition of cardoon seed oil; rather, it aims at simplifying the illustration of the concept. Codes refer to marine biodegradation tests.

**Figure 4 polymers-15-01536-f004:**
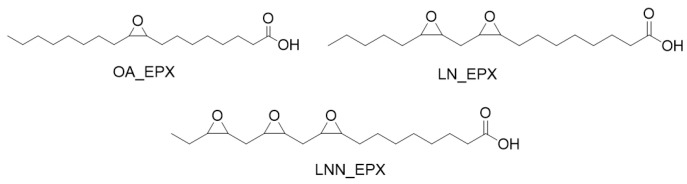
Structures of the products obtained from the enzymatically catalyzed epoxidation of oleic, linoleic, and linolenic acids. Codes refer to marine biodegradation tests.

**Figure 5 polymers-15-01536-f005:**
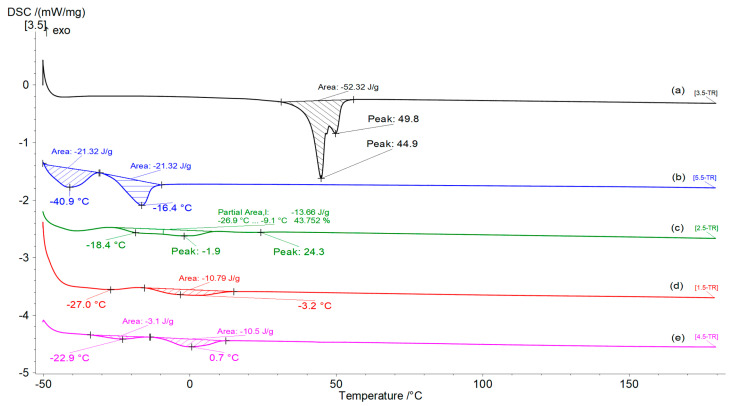
DSC thermograms of the samples (**a**) (BDO-AA)n, (**b**) (Gly-AZA)n, (**c**) ESO-BDO, (**d**) ECO-SORB, and (**e**) EPX-OIL.

**Figure 6 polymers-15-01536-f006:**
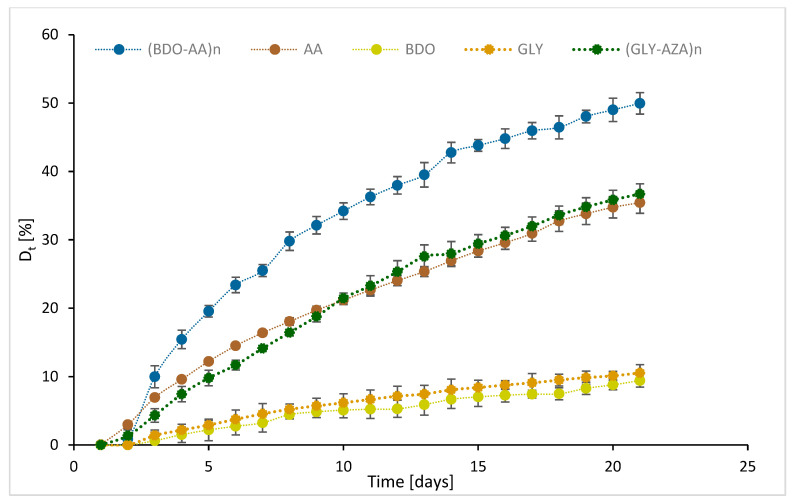
Degree of degradation of the oligoesters after 21 days of incubation in a marine environment; the data were normalized by subtracting the values of the control samples.

**Figure 7 polymers-15-01536-f007:**
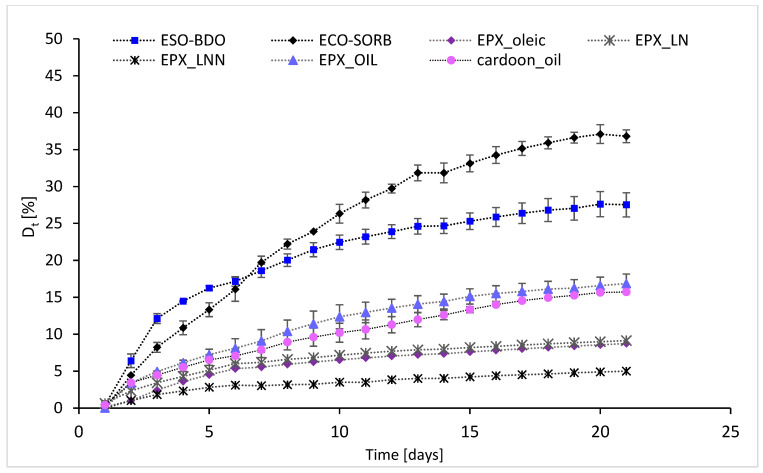
Degree of degradation of the epoxy cardoon oils and their derivatives after 21 days of incubation in a marine environment; the data were normalized by subtracting the values of the control samples.

**Table 1 polymers-15-01536-t001:** Conversions and medium molecular weight values of the enzymatically synthesized polyesters.

Sample	BDO-AA	Gly-AZA [27]
Conversion AA/AZA * (%)	94	96
M_n_ ** (g mol^−1^)	683	926
M_w_ ** (g mol^−1^)	729	945
Đ **	1.06	1.02

* Calculated based on NMR spectra; ** calculated based on micro-TOF spectra.

**Table 2 polymers-15-01536-t002:** Characterization of the derivatives of the cardoon seed oil.

Sample	Oxirane Number (go_2_/100g_sample_)	Hydroxyl Number (g_KOH_/100g_sample_)
ECO	6.7	-
ECO-BDO	0.07	188.1
ECO-SRB	0.1	178.1

**Table 3 polymers-15-01536-t003:** Experimental conditions for the epoxidation of pure oleic, linoleic, and linolenic fatty acids. Reactions were performed in the presence of immobilized CaLB (Novozyme 435) on a 2 g scale at 50 °C. The molar ratio between C=C bonds and H_2_O_2_ = 1:2.

Substrate	Amount of Enzyme (U/g_substrate_)	Reaction Time (h)	Conversion * C=C bond (%)
Oleic acid	225	2	>99
Linoleic acid	450	2	95
Linolenic acid	450	3	98

* Determined by ^1^H NMR.

**Table 4 polymers-15-01536-t004:** Mass losses over different temperature ranges of the oligoester and plasticizer.

Compound	Mass Loss (%)			
	20–200 °C	20–300 °C	20–400 °C	20–500 °C
(BDO-AA)n	1.17	4.15	59.22	98.74
ECO-BDO	4.22	19.80	67.87	99.98
ECO-SRB	6.93	25.96	69.31	99.10

**Table 5 polymers-15-01536-t005:** DSC parameters of the oligoesters and plasticizers.

Compound	T_i_ (°C)	T_f_ (°C)	ΔT (°C)	T_m_ (°C)	ΔH_m_ (J/g)	T_g_ (°C)
(BDO-AA)n	27.6	48.1	20.5	44.9	51.39	n.d.
(GLY-AZA)n	−29.2	9.4	19.8	−16.4	21.52	n.d.
ECO-BDO	−8.8	9.1	17.9	−1.9	13.66	−12.7
ECO-SORB	−1.3	13.5	14.8	−3.2	10.79	−16.0
EPX-OIL	−12.8	13.6	26.4	0.7	10.50	−11.2

**Table 6 polymers-15-01536-t006:** The BOD and D_t_ values obtained after 5, 10, and 21 days of degradation in a marine environment, along with the ThOD values of the monomers and oligoesters considered for biodegradation.

No	Sample	ThOD (mg/mg)	BOD_5_ (mg/L)	D_t5_ (%)	BOD_10_ (mg/L)	D_t10_ (%)	BOD_21_ (mg/L)	D_t21_ (%)
1	BDO	13.62	30.10	2.21	69.50	5.10	128.90	9.43
2	AA	6.38	28.11	12.23	48.70	21.19	81.90	35.44
3	GLY	5.30	15.50	2.92	32.80	6.18	55.95	10.55
4	(BDO-AA)n	1.32	25.95	19.55	45.40	34.20	66.30	49.95
5	(GLY-AZA)n	1.31	12.90	9.80	28.25	21.45	48.35	36.71
6	ESO-BDO	1.66	27.10	16.25	37.90	22.44	48.65	27.53
7	ECO-SORB	1.05	14.40	13.33	28.45	27.53	41.85	36.81
8	EPX_OIL	2.39	14.60	6.13	29.65	12.37	40.40	16.85
9	Cardoon_oil	2.69	14.90	5.52	27.55	10.20	42.25	15.72
10	EPX_oleic	7.21	34.55	4.51	50.60	6.53	67.25	8.74
11	EPX_LN	6.30	34.30	5.12	48.60	7.15	61.80	9.14
12	EPX_LNN	5.51	17.50	2.81	22.80	3.49	31.80	5.00

## Data Availability

Not applicable.

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
