# Peer review of "Understanding Marine Biodegradation of Bio-Based Oligoesters and Plasticizers"

_polymers, 2023, doi:10.3390/polym15061536_

Round 1

Reviewer 1 Report

Here are some comments

1.       Please provide reasons of choosing two oligoesters-poly(butyleneadipate) and poly(glycerolazelate). For examples, are they the most abundant or representative polyseters found in marine context?

2.       Some critical results such as DSC curves and NMR should be included in the main manuscript.

Author Response

Dear Reviewer,

Thank you for the kind suggestions. We revised the manuscript according to your comments.
Please, see the specific sections below.

  1. Please provide reasons of choosing two oligoesters-poly(butyleneadipate) and poly(glycerolazelate). For examples, are they the most abundant or representative polyesters found in marine context?

 The 2 aliphatic and linear oligoesters were selected by two main reasons: due to their potential applications as ingredients in cosmetic products and because they are both synthesized starting for industrially produced bio-based materials. Compared to their correspondent products obtained by chemical route the potential applications of the enzymatic synthesized products would be different and an evaluation of their behavior in marine environment was mandatory. As a matter of fact, their final use make their recovery and recycling unfeasible. Finally, a wealth of information (experimental and computational) were already gathered on their synthesis, structure and biotic/abiotic degradation. The following sentences were included to elucidate the concept (lines 268-297):

These oligoesters were selected not only for their potential industrial relevance but also because they have been already object of extensive investigations concerning their synthesis, structural properties and also their degradation. Previously we have re-ported the synthesis of oligoesters of poly(glycerolazelate) [27] for their potential use in dermatological and cosmetic formulations, due to the pharmacological properties of azelaic acid [29]. Therefore, these oligoesters are subjected to be washed off and col-lected in the wasterwater to reach, ultimately, the sea.

The fate of poly(butileneadipate) is similar, since it cannot be recovered nor recy-cled do to its final use as ingredient of adhesive and coating formulations.  Indeed, the synthesis poly(butileneadipate) was the first enzymatic polycondensation transferred on the industrial scale already in the ‘90s by Baxenden Chemicals (UK) for the pro-duction, later dismissed, of highly regular structures of polymers used for these ap-plications [30]. More recently, we have investigated the same enzymatic polyconden-sation but using recyclable covalently immobilized enzymes in thin film systems [31], also on pilot scale [32], to overcome the technical limitations observed  by Binns and co-workers. [30]. Our previous studies on the enzymatic synthesis of both poly(butileneadipate) and poly(glycerolazelate) indicate that the processes are poten-tially scalable, since the biocatalyst can be efficiently removed after the synthesis of short pre-polymers. As a matter of fact, even in the absence of the enzyme, the elonga-tion of the polyesters can be effectively pushed thermodynamically, by the removal of the water produced in the esterification.

Notably, previous computational simulations shed light on the ability of different hydrolases to recognize poly(glycerolazelate) as a substrate both in synthetic and hy-drolytic reactions [27]. Being the synthesis of the ester bond a reversible reaction, these oligoesters can in principle undergo a hydrolytic degradation catalyzed by the same enzymes used for their synthesis, namely triacylglycerol hydrolases, which are quite ubiquitous in nature. Furthermore, these oligoesters (Table 1) can be also considered as simplified models of polymers usable for formulating bio-based plastics. of low mo-lecular weight Their low molecular weight (between 683 and 926 g mol-1) allows to neglect factors such as shape and thickness of samples while focusing the attention only on specific chemical and structural factors at molecular level.

  1. Some critical results such as DSC curves and NMR should be included in the main manuscript.

Thank you for the comment. We included all the DSC thermograms to the main text (as Figure  5), and we revised the NMR assignation in the supplementary materials, according to the literature. Because of the NMR of the compounds have been previously reported, we believe the spectra can be maintained in the supplementary material. We updated the manuscript with more information about NMR in the experimental section. Please, kindly see lines 197-201.

Reviewer 2 Report

The study has results that could be of interest to the scientific community and the discussion is interestingly made. However, prior to being published in Polymers, I have some suggestions that could help the author give more scientific quality to their work.

1-      I would recommend the authors use some shorter keywords as they used long ones. This might help give higher repercussions to their work and make it found easily.

2-   The introduction section is supported by a low amount of references, as only 9 are used in the whole section. For instance, some of them are required in paragraph from line 63 to 75 and from line 76 to 85.

3-      Please, give more information about references 1, 2, 3 and 6.

4-     In the enzymatic activity assay, a further explanation about the essay is expected.

5-      Figure 6 plotted 7 different line-symbols but only 5 of them are collected in the legend. So, please make them different in the figure are collect them in the legend.

6-    Furthermore, samples in table 6 have different nomenclatures than those used in Figures 5 and 6. Please, make them coincident.

7-      Supplementary figures need of legend to be better understood.

Author Response

Dear Reviewer,

Thank you for the kind suggestions. We revised the manuscript according to your comments.
Please, see the specific sections below.

1 I would recommend the authors use some shorter keywords as they used long ones. This might help give higher repercussions to their work and make it found easily.

Thank you for the comment. We revised the keywords as follows:  Bio-based polyesters, bio-plasticizers, marine biodegradation, biocatalysis, epoxidized oil, epoxidized fatty acids, cardoon oil.

2 The introduction section is supported by a low amount of references, as only 9 are used in the whole section. For instance, some of them are required in paragraph from line 63 to 75 and from line 76 to 85.

Thank you for the suggestion. We added the references that were missing. Now the introduction includes 20 references.

3 Please, give more information about references 1, 2, 3 and 6.

We added the information to the references you’ve highlighted.

4 In the enzymatic activity assay, a further explanation about the essay is expected.

We updated the section adding more details about the enzymatic activity assay. Please, kindly see lines 127-135.

5 Figure 6 plotted 7 different line-symbols but only 5 of them are collected in the legend. So, please make them different in the figure are collect them in the legend.

We updated the image with the right legend.

6 Furthermore, samples in table 6 have different nomenclatures than those used in Figures 5 and 6. Please, make them coincident.

The figures and tables now are reporting the same sample names.

7 Supplementary figures need of legend to be better understood.

We added the summary of all figure legends into the supplementary materials.

Round 2

Reviewer 2 Report

I think that the manuscript has gained scientific quality with the changes made by the authors